# Participation and Performance Analysis in Children and Adolescents Competing in Time-Limited Ultra-Endurance Running Events

**DOI:** 10.3390/ijerph17051628

**Published:** 2020-03-03

**Authors:** Volker Scheer, Stefania Di Gangi, Elias Villiger, Thomas Rosemann, Pantelis T. Nikolaidis, Beat Knechtle

**Affiliations:** 1Ultra Sports Science Foundation, 69130 Pierre-Bénite, France; volkerscheer@yahoo.com; 2Health Science Department, Universidad a Distancia de Madrid (UDIMA), 28400 Collado Villaba, Madrid, Spain; 3Institute of Primary Care, University of Zurich, 8091 Zürich, Switzerland; Stefania.DiGangi@usz.ch (S.D.G.); evilliger@gmail.com (E.V.); thomas.rosemann@usz.ch (T.R.); 4Exercise Physiology Laboratory, 18450 Nikaia, Greece; pademil@hotmail.com; 5Medbase St. Gallen Am Vadianplatz, 9001 St. Gallen, Switzerland

**Keywords:** boy, girl, ultra-endurance, running, ultramarathon

## Abstract

Ultra-endurance running is of increasing popularity in the adult population, mainly due to master runners older than 35 years of age. However, youth runners younger than 19 years of age are also competing in ultra-endurance events, and an increase has been observed in distance-limited events, but no data is available on time-limited ultra-endurance events in this age group. This study investigated participation and performance trends in time-limited ultra-endurance races, including multi-day events, in runners younger than 19 years of age. Between the period 1990 and 2018, the most popular events recorded a total of 214 finishes (from 166 unique finishers (UF)) for 6-h events, 247 (212 UF) for 12-h events, and 805 (582 UF) for 24-h events, respectively. The majority of athletes originated from Europe and North America. Only a minority participated in multi-day events. Overall, speed increased with age, but the overall performance speed decreased across calendar years for 6- and 24-h events as participation numbers grew. In summary, in youth ultra-endurance runners, differences were observed regarding participation and performance across the different time-limited events, the age of the athletes and their country of origin

## 1. Introduction

Ultra-endurance running can be defined as running activities lasting longer than six hours [1,2,3]. These activities can be distance- or time-based, with typical time-based ultra-endurance events ranging from six hours to several days, the distance covered during this time period being recorded and ranked among competitors [4]. Popular time-based events in the adult population include races over 24 h and participation numbers have increased over the years, particularly among master athletes and women [3,5]. Men are generally faster than women, however, women have closed the gap in the last decade [6]. Most adult ultra-endurance athletes in 24-h events originate from Europe, mostly France and Germany [6]. Multi-day ultra-endurance events are also quite popular, with an exponential increase between the 1990s and 2010 [7]; however, competitor numbers are generally lower compared to other ultra-endurance races. Again, most finishers come from Europe, mainly France, the United Kingdom and Germany, followed by runners from the USA, Asia, Africa, Australia and South America [7]. Ultra-endurance races can be held in challenging and remote environments, and can include races in the heat, desert, cold or jungle environments [3,8,9,10,11]. Peak running performance has increased with increasing event duration in races lasting from hours to ten days in adults [4].

As outlined, the participation and performance trends of ultra-endurance running are well described in the adult population [12,13], as well as the performance differences among sexes [14,15]; however, very little is known about childhood participation in ultra-endurance running events and if, indeed, they should be participating in them at all [16]. For ultra-marathons, mainly the aspects of age [4,17] and nationality [18,19] have been investigated. One concern is that running and training for ultra-endurance distances at a young age can have acute negative effects on an immature or developing muscular skeletal system or another organ system. Other concerns may relate to the long-term negative health effects this could have; however, this is currently not known and has not been investigated [15].

Until recently there was only anecdotal evidence that children and adolescents participate in ultra-endurance events; however, one study was able to demonstrate that the participation of children and adolescents younger than 19 years old is a reality [3]. This study looked at participation trends among youth ultramarathon runners and described an exponential increase in participation in the last 20 years [16]. The most popular race distances were those of 100 km, followed by 50 km and 50 miles, with the majority of finishers being older boys between 16–18 years of age [20]. However, ultramarathon running is only distance based, defined as races over marathon distance (42.195 km), and is quite distinct to time-based or multi-day events. Youth ultra-endurance athletes first participated in multi-day events in the year 2000, with events ranging from two to eight days, with distances covering 81 to 293 km, respectively; however, less than 50 runners participated in these events in the last two decades [20].

To date, there are no data available on the country of origin of the ultra-endurance youth participants and similarly no data are available on their performance times. This is the first study examining participation and performances in time-limited ultra-endurance and multi-day events in youth runners. This is of practical interest to scientists, coaches and health care professionals, looking after youth ultra-endurance runners, to get a better understanding of participation trends and performance times. Our aim was, therefore, to examine participation numbers and trends, including countries of origin, race performance times and speeds and sex differences, in children and adolescents younger than 19 years of age in time-based ultra-endurance and multi-day running events. Our hypothesis was that participation numbers would increase over calendar years and in time, more boys than girls would participate in these events, and that the performance times from boys would be faster.

## 2. Materials and Methods 

### 2.1. Ethical Approval

The study was approved by the medical council (Ärztekammer Westfalen Lippe, Germany) and the University of Münster, Germany (Chairperson Prof Berdel, protocol number 2018-304-f-S).

### 2.2. Data Sampling and Data Analysis

All data were obtained from the Deutsche Ultramarathon Vereinigung (DUV) website where all the race results of ultramarathons are recorded (https://statistik.d-u-v.org/index.php). The DUV is the largest ultra-running database worldwide, containing more than 5.8 million performances of more than 1.4 million runners in approximately 60,000 ultramarathon events and is widely used to gain insights in participation and performance trends in ultra-running (www.ultra-marathon.org/) [3,16,17]. A computer script was written to retrieve a list for every event recorded on the website. Each event’s web page was then read by the script to extract the complete data table available. The script compiled all that data into one large Excel file, which was our starting point for further manual filtering of relevant information. We extracted data of time-limited races (i.e., 6, 8, 12, 24, 48, 72 h, 6 and 8 days) from 1990 to 2018.

The following variables were extracted: year of race, race distance/duration, name of the race, race performance (km or miles), name of athlete, year of birth, nationality of athlete, and sex of athlete. Running speed (km/h) was calculated from the performance and duration variables. Age was obtained by subtracting the year of birth from the year when the race was held. Continent variable was defined from the nationality of the athletes.

### 2.3. Statistical Analysis

The outcome was the running speed (km/h). Information for all races: number of observations, mean (SD) and minimum and maximum of speed (km/h) is provided in Table 1. For the main analyses, time-limited races of 8, 48, 72 h, 6 and 8 days were excluded due to insufficient data (< 100 observations). Descriptive statistics were presented as means (SD = standard deviations) by sex, age groups, continents and time groups. The age groups were 10–13, 14–15, 16–17 and 18 years. The continent groups, with reference to the nationality of the athletes, were: Africa, Asia, Central-South America, Europe, North-America, Oceania. When the number of observations of each continent group, within each race, was not greater than ten, continents were grouped together into other continents. To show a performance by a period of time, the calendar year of the race was grouped into time periods of 10 years. Age and calendar year were considered as continuous variables, in 1-year intervals, when defined as predictor variables for ultra-running speed. In fact, non-linear regression mixed models, with basis splines (BS), were performed to examine the time trend together with the effects of sex, age and continent on the speed time of each duration race. The mixed models were used to correct repeated measurements within runners (clusters) through the random effects of intercepts. The statistical models were specified as follows:
Ultra-running speed (Y) ~ [Fixed effects (X) = BS(Year, df = 3) + BS(Age, df = 3) *sex + continent + [random effects of intercept=runners]
where BS(Year, df=3) and BS(Age, df = 3) are three degrees of freedom (df) basis splines changing with calendar year and age, respectively; BS(Age, df = 3) * sex denote the age–sex interaction term. Different analyses were performed, one for each duration (6, 12, and 24 h). The interaction term age–sex was significant and considered only in the 24-h events. In the 6- and 12-h race analyses, a linear term on year, instead of a spline term, was considered. Results of the regression models were presented as estimates and standard errors. Statistical significance was defined as p < 0.05. All statistical analyses were carried out with R, R Core Team (2016). R: A language and environment for statistical computing. R Foundation for Statistical Computing, Vienna, Austria. URL https://www.R-project.org/. The R packages ggplot2, lme4, and lmerTest were used, respectively, for data visualization and for the mixed models.

## 3. Results

Between 1990 and 2018, the total number of observations, over 6-, 12- and 24-h races was *n* = 214, *n* = 247, and *n* = 805 records, respectively. Instead, the number of individual finishers was, respectively, *n* = 166, *n* = 212 and *n* = 582. We observed that the percentage of children aged 10–13 years was relatively high. In fact, in 24-h races, the majority were 10–13 years, 376 (46.7%). Instead, in 6-h races, the majority of finishers, 81 (37.9%), were 18 years and in the 12-h race, the majority were 16-17 years, 83 (33.6%).

In the 6- and 24-h races, 146 (68.2%) and 441 (54.8%), respectively, came from Europe but in the 12-h races, the majority of 130 (52.6%) came from North America. In time-limited races, however, no participation was recorded before 1990 and the vast majority participated within the last 8 years. The number of observations and the average performance by sex, age groups, continent and time groups are reported for distance races, respectively, in Table 2. In Figure 1, Figure 2 and Figure 3, the participation (%) and average performances (km/h) by nationality for each time-limited race are reported. Table 3 describes the results of the statistical models, as described in the methods section. Boys were significantly faster than girls only in the 12-h race. There were no significant differences between North America, Europe and other continents.

For time-limited races (Figure 4), a time effect was not significant in 6- and 12-h events but it was significant in 24-h races, where running speed decreased over time. In all time-limited races, running speed increased across age groups (Figure 5) and this trend was different between boys and girls.

## 4. Discussion

This is the first study that examined the participation and performances in time-limited ultra-endurance events and multi-day events in youth runners. The aim of the present study was to investigate the age-related participation and performance trends of children and adolescent ultra-endurance runners, younger than 19 years of age, in time-limited events. Our hypothesis was that participation numbers would increase over calendar years and time, more boys than girls would participate in these events, and that performance times from boys would be faster. The main findings were (*i*) an increase in the number of ultra-endurance participation over time and across races and sexes, (*ii*) performance differences between boys and girls, with boys being significantly faster than girls only in the 12-h race, (*iii*) differences between running speed across age groups and continents and (*iv*) variations in running speed over the years and different age–sex trend in 24-h events.

### 4.1. Participation Trends

An exponential increase in participation numbers among youth ultra-runners was observed in the last 30 years. The most popular time-limited race competitions were 24, 12 and 6 h long; however, participation numbers were considerably smaller. The majority of finishers belonged to the older age groups (16–18 years of age) and were mostly male. Our findings confirm previously observed participation trends [16]. In time-limited races, most of the finishers in the 6- and 24-h races came from Europe but, for 12-h races, most originated from North America. A study investigating the sex difference in 24-h adult ultramarathoners showed that most of the starters originated from Europe, mainly France and Germany [6]. Only a small minority participated in multi-day events.

### 4.2. Differences in Age Regarding Duration of Races

The percentage of runners aged 10-13 years was rather high in time-limited races. The existing literature for adult ultramarathoners investigated the age trends only for distance-limited races [2,21] but it made no comparison between the different kinds of ultra-endurance events. In the 6-h races, the majority of the finishers were 18 years-old, whereas in the 12-h races, the majority of the competitors were between 16–17 years of age. For adult ultramarathoners, it has been reported that the age of peak running performance increased with race duration with time-limited events ranging from 6 h to 10 days [4]. Obviously, there is a difference between youth and adult ultra-endurance runners, where youth athletes seem to be older in longer time-limited races, in contrast to adult runners, where the opposite was found.

### 4.3. Analysis among Different Races and Sexes

Boys were faster than girls in the 12-h races, but not in the 6- and 24-h races. This is hardly surprising, taking the physiological differences during and after maturation into account. For adult ultramarathoners, men were generally faster than women in time-limited ultramarathons (e.g., 24-h races) [6]. Women, however, have closed the gap with men in the last decade [6,17,22,23]. The mean ultra-endurance running speed was generally faster in boys than girls across all time limited ultra-endurance races; however, this was only significant for the 12-h races. Younger girls, under the age of 15 years, were faster than boys in the 6-h races. One possible explanation for the latter finding may be that girls mature physiologically earlier than boys and this may have given them an advantage at this race distance. Another explanation may be that there were far fewer girls participating at this particular race distance and that they may have been better prepared for this event.

### 4.4. Analysis in Running Speed Across Age Groups

In all time-limited races, running speed increased across age groups and was different between boys and girls in 24-h races, with boys being faster than girls. In other terms, boys and girls improved their running performance with increasing age in time-limited races. However, they were still far away from the age of peak ultramarathon performance which is generally achieved at ages beyond 35 years [2,4,21] and increases with increasing race duration in time-limited races [4]. It is well-known that the age of peak performance in endurance sports increases with increasing length or duration of the endurance performance [24].

### 4.5. Analysis in Running Speed over Years

The last important finding was that the running speed showed differences in the trend across calendar years for the different ultra-endurance events. A time effect was not significant in 6- and 12-h events but it was significant in 24-h races, where running speed decreased over time. Taken together, these youth ultra-runners were not able to improve their running performance in recent years, although in some races in earlier years their running speed was higher than in recent years. Obviously, there is a general trend in long-distance races that running speed decreased in recent years, and this may be related to the general increase in participation numbers, but not necessarily an increase in faster or more elite runners, that would increase the overall performance times.

### 4.6. Limitations

The analyzed data originate from one database (DUV-www.ultra-marathon.org/). We recognize that there are several other national databases; however, the DUV is the largest ultra-running database worldwide and has been used widely to address similar research questions in the adult population. Inaccuracies in reporting or missing datasets are possible in such a large database. For the analysis of particular national races, data from the race websites could be analyzed for future studies. Several races have been grouped together, without taking into account the specific ambient, environmental or terrain particularities, which can have an impact on the average performance time. To address this, we recommend analyzing specific races in future.

### 4.7. Practical Applications

In last decade, a large increase in the number of finishers and annual races of ultra-marathons has been observed. Following this trend, the number of children and adolescents competing in these races has increased, too. Consequently, strength and conditioning trainers could face new challenges when working with children and adolescent ultramarathoners, since the existing knowledge is based mostly in studies on adult athletes. The present study added valuable practical information in the existing literature with regards to trends in the participation and performance of children and adolescent ultramarathoners. For instance, these trends varied by country; thus, the findings would especially be of practical interest for strength and conditioning trainers working in countries with increased participation of these age groups in ultramarathons. Future studies may investigate the same trends in distance-limited ultra-marathons such as 100 km and 100 miles.

## 5. Conclusions

Comparing time-limited races from 6 h to 8 days, it was concluded that ultramarathoners younger than 19 years of age participated mostly in 6-, 12- and 24-h races, and the majority of these athletes originated from Europe and North America. Only a minority participated in multi-day events. Overall, speed was faster in the older rather than in the younger athletes. Finally, the overall speed mostly decreased across calendar years as participation numbers grew.

## Figures and Tables

**Figure 1 ijerph-17-01628-f001:**
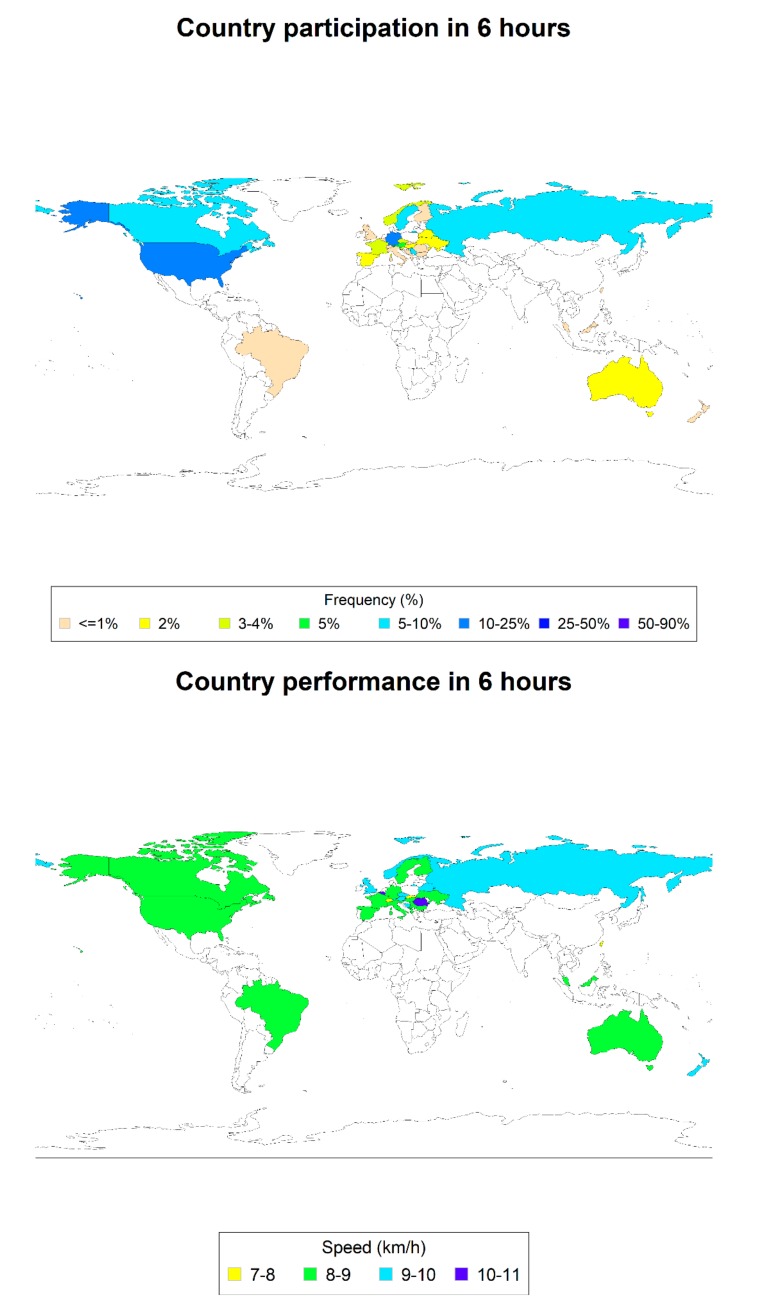
Participation (%) and average performance (km/h) by nationality in 6-h races.

**Figure 2 ijerph-17-01628-f002:**
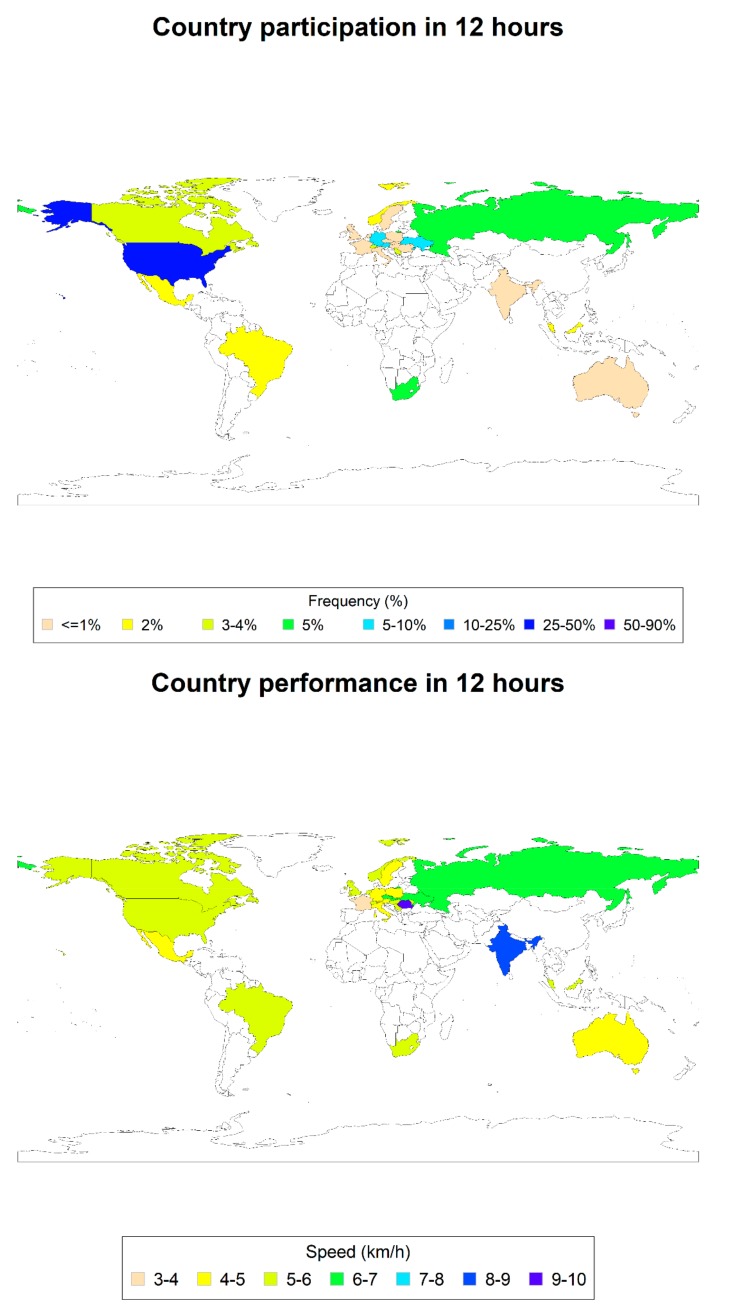
Participation (%) and average performance (km/h) by nationality in 12-h races.

**Figure 3 ijerph-17-01628-f003:**
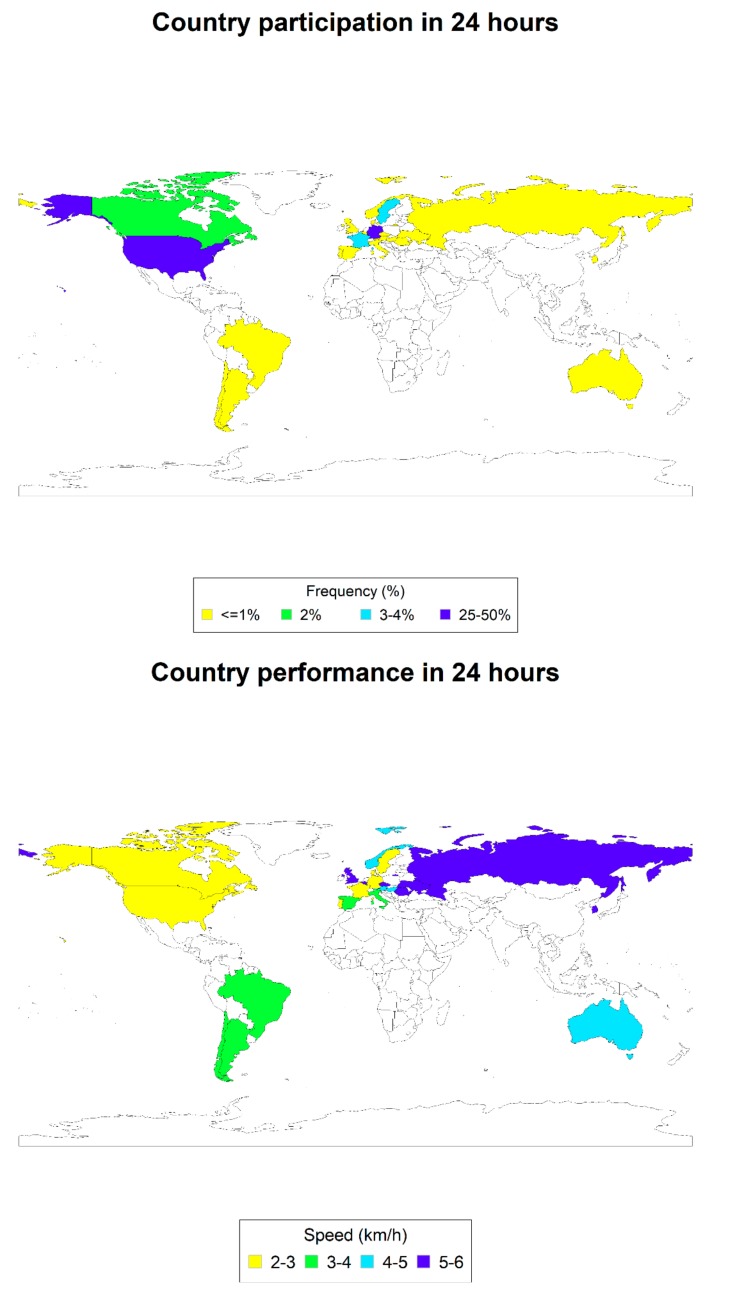
Participation (%) and average performance (km/h) by nationality in 24-h races.

**Figure 4 ijerph-17-01628-f004:**
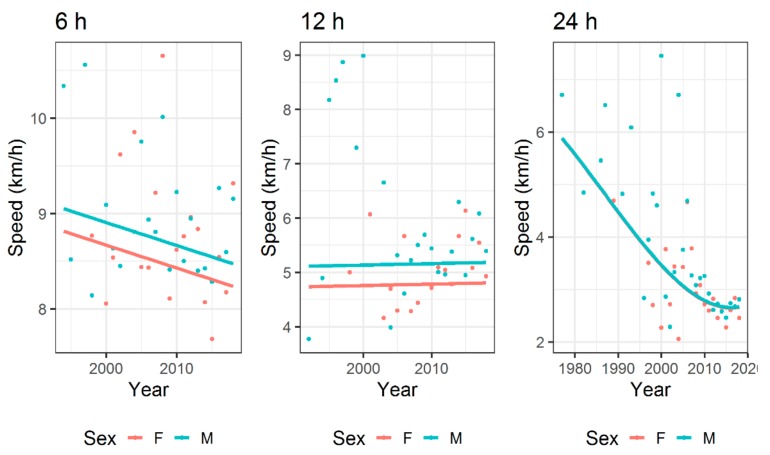
Running speed across years for time-limited ultra-endurance events for 6, 12 and 24 h. Fitted values=line, points=observed mean values.

**Figure 5 ijerph-17-01628-f005:**
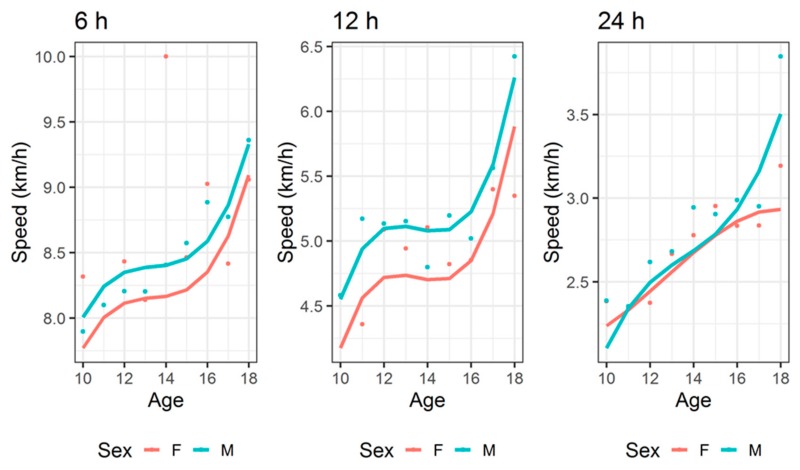
Running speed across age groups for time-limited ultra-endurance events for 6, 12 and 24 h. Fitted values=line, points=observed mean values.

**Table 1 ijerph-17-01628-t001:** Ultra-endurance performance—speed km/h by duration. Mean (SD) and minimum (Min), maximum (Max) were reported.

Duration		Running Speed (km/h)		
	*N*	Mean (SD)	Min	Max
6 h	214	8.84 (1.06)	7.50	12.83
8 h	68	6.97 (0.92)	5.68	10.24
12 h	247	5.39 (1.36)	3.75	10.42
24 h	805	2.83 (1.00)	1.87	8.00
48 h	46	2.40 (1.06)	0.94	4.37
72 h	50	1.54 (0.72)	0.66	3.24
6 days	13	1.57 (0.77)	0.45	3.37
8 days	7	1.72 (0.35)	1.15	2.30

**Table 2 ijerph-17-01628-t002:** Mean ultra-endurance running speed in km/h and (SD): duration of races (6, 12 and 24 h) by sex, age, country (continent) and calendar year groups. Africa, Asia, Central-South America, Oceania, due to small sample size, were combined into “Other” group.

Duration		6 h, *N* = 214		12 h, *N* = 247		24 h, *N* = 805	
Age	Sex	*N*	Mean (SD)	N	Mean (SD)	*N*	Mean (SD)
10–13	F	5	8.23 (0.34)	24	4.94 (0.93)	136	2.45 (0.56)
	M	18	8.16 (0.51)	22	5.05 (0.98)	240	2.53 (0.77)
14–15	F	7	8.68 (0.66)	18	4.91 (1.10)	84	2.84 (0.89)
	M	32	8.47 (0.93)	36	5.01 (1.10)	101	2.93 (0.86)
16–17	F	17	8.52 (0.91)	23	5.21 (1.08)	55	2.83 (0.94)
	M	54	8.82 (0.91)	60	5.34 (1.32)	104	2.97 (0.93)
18	F	20	9.09 (0.78)	17	5.35 (0.80)	26	3.19 (1.01)
	M	61	9.33 (1.31)	47	6.42 (1.79)	86	3.85 (1.56)
**Continent**	**Sex**	***N***	**Mean (SD)**	***N***	**Mean (SD)**	***N***	**Mean (SD)**
North America	F	11	8.65 (0.96)	53	5.11 (1.02)	106	2.90 (0.94)
	M	50	8.71 (0.91)	77	5.51 (1.34)	246	3.03 (1.07)
Europe	F	35	8.77 (0.81)	25	5.07 (0.98)	162	2.50 (0.63)
	M	111	8.96 (1.20)	68	5.54 (1.70)	279	2.77 (1.08)
Other	F M	3 4	8.87 (0.75) 8.22 (0.53)	4 20	5.02 (0.79) 5.62 (1.43)	6 6	3.60 (0.95) 4.21 (0.71)
**Year**	**Sex**	***N***	**Mean (SD)**	***N***	**Mean (SD)**	***N***	**Mean (SD)**
1990–1999	F	1	8.77	1	5.00	2	3.10 (0.57)
	M	8	9.64 (1.26)	6	6.92 (2.10)	17	4.03 (1.79)
2000–2009	F	19	8.87 (0.77)	16	4.76 (0.97)	23	3.26 (1.05)
	M	52	8.99 (1.26)	43	5.35 (1.69)	69	3.50 (1.52)
2010–2018	F	29	8.67 (0.88)	65	5.18 (0.99)	248	2.61 (0.75)
	M	105	8.75 (1.00)	116	5.53 (1.36)	439	2.73 (0.82)

**Table 3 ijerph-17-01628-t003:** Regression analysis (mixed model) of ultra-endurance events (6, 12 and 24 h). Estimates and standard errors (SE) of fixed effects are reported. *p*-values ranges are marked with asterisks (see note). Smoothing terms, basis splines (BS), are denoted with BS(x) t, where x = year, age; t = 1,2,3.

Predictor	Time-Limited Races		
	6 h	12 h	24 h
**Age**			
BS(Age)1	0.842 (0.940)	1.423 (1.173)	0.227 (0.382)
BS(Age)2	−0.226 (0.538)	−0.584 (0.628)	0.708 *(0.331)
BS(Age)3	1.323 **(0.482)	1.708 **(0.590)	0.696 ***(0.203)
**Sex** = M (ref=F)	0.236 (0.185)	0.377 *(0.189)	−0.134 (0.167)
**Age:Sex interaction terms**			
BS(Age)1:SexM			0.551 (0.475)
BS(Age)2:SexM			−0.402 (0.402)
BS(Age)3:SexM			0.703 **(0.243)
**Year**	−0.024 (0.014)	0.003 (0.017)	
BS(Year)1			−1.349 (1.210)
BS(Year)2			−3.417 ***(0.612)
BS(Year)3			−3.215 ***(0.677)
**Continent (ref. North America)**			
Europe	0.142 (0.173)	−0.140 (0.196)	−0.115 (0.069)
Other		−0.238 (0.328)	0.395 (0.296)
Constant	56.062 *(27.523)	−1.086 (33.910)	5.495 ***(0.684)
Observations Runners	207 159	247 212	805 582

*Notes:* * *p* < 0.05; ** *p* < 0.01; *** *p* < 0.001.

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
