# Peer review of "Participation and Performance Analysis in Children and Adolescents Competing in Time-Limited Ultra-Endurance Running Events"

_ijerph, 2020, doi:10.3390/ijerph17051628_

Round 1

Reviewer 1 Report

Introduction

In the introduction section, the climatic characteristics of the events. are reported; however, in the analysis and discussion of the results, there is no description of these conditions.

What does this study add to the one published by Scheer et al. 2019?

The practical application of the study should be reported here.

Methods

Only one database was used: All data were obtained from the website of DUV (Deutsche Ultramarathon Vereinigung); however, there are many other national registries that could be used.

Results

In my opinion, the information reported in the figures is low. Would it be possible to combine or include further analysis?

The number of points (data) in Figure 4, does not correspond with the sample reported above.

The data interpretation and the conclusions are only partly supported. The main work is to adapt the conclusions to the results and develop a regime based on the own actual findings combined with actual results of scientific literature

Minor comments:

Add extra space in p.2, L53 and P10, L160 y 161.

P2. L75, space after “e.6”

Add final stop in p.2, L68.

Author Response

Reviewer 1

Introduction

In the introduction section, the climatic characteristics of the events. are reported; however, in the analysis and discussion of the results, there is no description of these conditions.

Reply: Thank you. These are general conditions where ultra-endurance races can take place and demonstrate the additional environmental demands (apart from the distance) athletes sometimes compete in. The data include a multitude of races and specific conditions are not know and therefore further analysis cannot be performed. This aspect has been added to the limitation section and we have clarified the statement in the introduction.

What does this study add to the one published by Scheer et al. 2019?

Reply: Thank you. Our previous study was the first of its kind to examine youth participation in ultra-endurance events and gave a general overview of participation numbers. This current study is much more targeted and the analysis not only incudes the participation trends of time limited ultra-endurance events but is also the first study to report on performance trends. This is unique and the first study to examine this in youth ultra-endurance athletes. This has been added to the introduction.

The practical application of the study should be reported here.

Reply: Thank you. We have added the practical application in the introduction.

Methods

Only one database was used: All data were obtained from the website of DUV (Deutsche Ultramarathon Vereinigung); however, there are many other national registries that could be used.

Reply: Thank you. The DUV database is the largest ultra-running database worldwide, containing more than 5.8 million performances of more than 1.4 million runners in approximately 60.000 ultramarathon events and is widely used to gain insights in participation and performance trends in ultra-running. We have added a comment into the methods section and limitations.

Results

In my opinion, the information reported in the figures is low. Would it be possible to combine or include further analysis?

Reply: Thank you. The intention of the figures is to easily provide a visual context of the results of participation and performance trends based on countries and race distance. We believe this is a vital part of visualising the results and provides clarity and reviewer 3 seems to concur as well. We have therefore opted to retain the figures in its original version.We decided to show Figures 1-3, instead of Tables because we think that they help the reader to have a quick idea of the differences, in terms of participation and performances within each continent. Anyway, the essential summary statistics (by continent and other factors) were reported in Table 2. Moreover, we showed visually the effects of the statistical models, fully reported in Table 3. We acknowledge that maybe other analyses could have been done. But for the purposes of this study, we think they are sufficient.

The number of points (data) in Figure 4, does not correspond with the sample reported above.

Reply: We thank the reviewer very much for having spotted this. We didn’t specify that the points in Figure 4 are the mean values of the observations. We have clarified it in the Figures Caption.

The data interpretation and the conclusions are only partly supported. The main work is to adapt the conclusions to the results and develop a regime based on the own actual findings combined with actual results of scientific literature

Reply: We revised conclusions to match results.

Minor comments:

Add extra space in p.2, L53 and P10, L160 y 161.

Reply: Thank you. We changed as requested.

P2. L75, space after “e.6”

Reply: Thank you. We changed as requested.

Add final stop in p.2, L68.

Reply: Thank you. We changed as requested.

Reviewer 2 Report

Introduction: Would be helpful to readers to include some of the data that exists for the adult population to use as a reference when comparing the data you present in the results to aid in comparing. The sources are listed for readers to find themselves but this may aid in the overall readability and interest to the reader. Mainly in areas of lines 50-53 for introduction.

Methods: No comments. Very straightforward.

Results: Lines 114-116. Difficult to follow this section and what numbers refer to which duration of event. Please revise.

Table 2. Continent Section: North America should align with the F for female to match up with other sections for age and year.

Figures 1, 2, 3. Very hard to visualize. Consider changing color and altering sizing of figures to make it easier to visualize the figure. Also, the lines need to be more defined to make observations regarding each country. Very grainy pictures.

Table 3. Condense to fit on one page rather than running over to next page.

Discussion: Lines 177 & 178 - add space between "6hours" and "12hours"

4.3 - Statement of boys were faster in 12 hours but not 6 hours and 24 hours then followed by hardly not surprising. The findings are somewhat inconsistent since they are not consistent across longer or shorter duration events so this statement is misleading. Consider revising to address why the differences were not consistent between the different duration events between sex.

4.5 - Link the fact of decreased running speeds being due to increased participation as stated in section 4.1. Normally, you would expect as years go on, performance will increase with training quality and preparation improving. As you highlight in conclusion, this needs to be addressed but it has yet to increase in popularity for performance to improved, just participation has increased but not necessarily participation of better performers.

Author Response

Reviewer 2

Introduction: Would be helpful to readers to include some of the data that exists for the adult population to use as a reference when comparing the data you present in the results to aid in comparing. The sources are listed for readers to find themselves but this may aid in the overall readability and interest to the reader. Mainly in areas of lines 50-53 for introduction.

Reply: we inserted ‘For ultra-marathons, mainly the aspect of age [4,17] and nationality [18,19] have been investigated

Methods: No comments. Very straightforward.

Results: Lines 114-116. Difficult to follow this section and what numbers refer to which duration of event. Please revise.

Reply: Thank you. We agree with the reviewer and we have rephrased it.

Table 2. Continent Section: North America should align with the F for female to match up with other sections for age and year.

Reply: Thank you. We thank the reviewer for having remarked this and we have corrected it.

Figures 1, 2, 3. Very hard to visualize. Consider changing color and altering sizing of figures to make it easier to visualize the figure. Also, the lines need to be more defined to make observations regarding each country. Very grainy pictures.

Reply: Thank you. We thank the reviewer for having remarked this and we have changed the Figures.

Table 3. Condense to fit on one page rather than running over to next page.

Reply: Thank you. We agree with the reviewer and we have edited it accordingly.

Discussion: Lines 177 & 178 - add space between "6hours" and "12hours"

Reply: Thank you. We changed as requested.

4.3 - Statement of boys were faster in 12 hours but not 6 hours and 24 hours then followed by hardly not surprising. The findings are somewhat inconsistent since they are not consistent across longer or shorter duration events, so this statement is misleading. Consider revising to address why the differences were not consistent between the different duration events between sex.

Reply: Thank you. We added “Mean ultra-endurance running speed was generally faster in boys than girls, across all time limited ultra- endurance races, however this was only significant for the 12-hour races. Younger girls under the age of 15, were faster than boys in the 6-hour races. One possible explanation for the latter finding may be that girls mature physiologically earlier than boys and this may have given them an advantage at this race distance. Another explanation may be that there were far fewer girls that participated at this particular race distance and that they may have been better prepared for this event.”

4.5 - Link the fact of decreased running speeds being due to increased participation as stated in section 4.1. Normally, you would expect as years go on, performance will increase with training quality and preparation improving. As you highlight in conclusion, this needs to be addressed but it has yet to increase in popularity for performance to improved, just participation has increased but not necessarily participation of better performers.

Reply: Thank you. We added “Obviously, this is a general trend in long-distance races that running speed decreased in recent years, and this may be related to the general increase in participation numbers, but not necessarily an increase in faster or more elite runners, that would increase the overall performance times.”

Reviewer 3 Report

This study aimed to investigate the age-related participation and performance trends of children and adolescent ultra-endurance runners younger than 19 years of age in time-limited events. The results of the study showed an increase in the number of ultra-endurance participation over time and across races and sexes; performance differences between boys and girls, with boys being significantly faster than girls, only in 12 hours race; differences between running speed across age groups and continents and; variations in running speed over the years and different age-sex trend in 24 hours.

The research question is relevant. Regarding the introduction, we believe that it offers an adequate contextualization about the previous empirical and theoretical literature related to the object of study. Nevertheless, it is also important to indicate, in a more developed way, the value or potential meaning of the problem area. Also, it would be interesting for the reader to present and substantiate the hypotheses. The methods used are appropriate to achieve the objective of the study. Also, the presentation of the results in the textual form is performed in an appropriate and noticeable way for the reader. In addition, the figures and tables show the essential data, not revealing duplicate data in the tables and text.

In Discussion, make sure this opening sentence really explains the uniqueness and novelty of this study. In addition, pay attention to the following: line 161 “(iii) differences between running speed across age groups and continents”. According to the results, “there were no significant differences between North America, Europe and other continents” (line 126). The authors may also indicate the limitations of the study. Also, in discussing the limitations of the study, please make suggestions for future research regarding how these limitations could be addressed.

Lastly, some references do not agree with ACS style guide.

Author Response

Reviewer 3

This study aimed to investigate the age-related participation and performance trends of children and adolescent ultra-endurance runners younger than 19 years of age in time-limited events. The results of the study showed an increase in the number of ultra-endurance participation over time and across races and sexes; performance differences between boys and girls, with boys being significantly faster than girls, only in 12 hours race; differences between running speed across age groups and continents and; variations in running speed over the years and different age-sex trend in 24 hours.

The research question is relevant. Regarding the introduction, we believe that it offers an adequate contextualization about the previous empirical and theoretical literature related to the object of study.

Nevertheless, it is also important to indicate, in a more developed way, the value or potential meaning of the problem area.

Reply: Thank you. We have added further details into the introduction. “One concern of course is, that running and training for ultra-endurance distances at a young age can have acute negative effects on an immature or developing muscular skeletal system or another organ system. Other concerns may relate to the long-term negative health effects, this could have, however this is currently not known and has not been investigated. Up until recently there was only anecdotal evidence that children and adolescents participate in ultra-endurance events, however one study was able to demonstrate that participation of children and adolescents younger than 19 years is a reality.

Also, it would be interesting for the reader to present and substantiate the hypotheses.

Reply: Thank you: We have clarified the hypothesis in the introduction. Our hypothesis was that participation numbers would increase over calendar years and time, more boys than girls would participate in these events, and that performance times from boys would be faster.

The methods used are appropriate to achieve the objective of the study. Also, the presentation of the results in the textual form is performed in an appropriate and noticeable way for the reader. In addition, the figures and tables show the essential data, not revealing duplicate data in the tables and text. In Discussion, make sure this opening sentence really explains the uniqueness and novelty of this study.

Reply: Thank you. We have added the introductory sentence in the discussion to explain the uniqueness and novelty of our study. This is the first study that examined the participation and performances in time limited ultra-endurance events and multi-day events in youth runners. The aim of the present study was to investigate the age-related participation and performance trends of children and adolescent ultra-endurance runners younger than 19 years of age in time-limited events. Our hypothesis was that participation numbers would increase over calendar years and time, more boys than girls would participate in these events, and that performance times from boys would be faster.

In addition, pay attention to the following: line 161 “(iii) differences between running speed across age groups and continents”. According to the results, “there were no significant differences between North America, Europe and other continents” (line 126).

Reply: Thank you. We deleted continents

The authors may also indicate the limitations of the study. Also, in discussing the limitations of the study, please make suggestions for future research regarding how these limitations Reply: Thank you. We added a section ‘Limitations’ with: Data analysed originate from one database (DUV- reference). We recognise that there are several other, national databases, however the DUV database is the largest ultra-running database worldwide and has been used widely to address similar research questions in the adult population. Inaccuracies in reporting or missing data sets are possible in such a large database. For the analysis of particular national races, data from the race websites could be analysed for future studies. Several races have been grouped together, without taking into account the specific ambient, environmental or terrain particularities, which can have an impact in the average performance time. To address this, we recommend to analyse specific races in future.

Lastly, some references do not agree with ACS style guide.

Reply: Thank you. We revised references for style.

Round 2

Reviewer 1 Report

I thank the authors for the effort to respond to all my suggestions